# Microstructure and Mechanical Properties of Al-Si Alloys Produced by Rapid Solidification and Hot Extrusion

**DOI:** 10.3390/ma16155223

**Published:** 2023-07-25

**Authors:** Piotr Noga, Tomasz Skrzekut, Maciej Wędrychowicz

**Affiliations:** 1Faculty of Non-Ferrous Metals, AGH University of Science and Technology, A. Mickiewicza Av. 30, 30-059 Krakow, Poland; pionoga@agh.edu.pl (P.N.); skrzekut@agh.edu.pl (T.S.); 2Faculty of Mechanical Engineering, Institute of Materials and Biomedical Engineering, University of Zielona Góra, Prof. Z. Szafrana Street 4, 65-516 Zielona Góra, Poland

**Keywords:** Al-Si aluminum alloy, melt-spinning, shape factor, mechanical properties

## Abstract

The paper presents the results of tests of rapid solidification (RS) aluminum alloys with the addition of silicon (5%, 11%, and 20%). Casting by melt-spinning on the surface of an intensively cooled copper cylinder allowed to obtain a metallic material in the form of flakes, which were then consolidated in the process of pressing and direct extrusion. The effect of refinement on structural components after rapid solidification was determined. Rapidly solidified AlSi materials are characterized by a comparable size of Si particles, regardless of the silicon content, and the shape of these particles is close to spheroidal. Not only Si particles are fragmented, but also the Al-Si-Fe phase, which also changed its shape from irregular with sharp edges to regular and spherical. The melt-spinning process resulted in a fine-grained structure compared to materials obtained by gravity-casting and extrusion. The influence of the high-temperature compression test on the mechanical properties of rapidly solidified materials was analyzed, and the results were compared with those of gravity-cast materials. An increase in strength properties was found in the case of the AlSi5 RS alloy by 20%, in the case of AlSi11RS by 25%, and in the case of the alloy containing 20% Si by as much as 86% (tensile test). On the basis of the homogeneity of the particle distribution determined by the SEM method, it was found that rapid solidification is an effective method of increasing the strength properties and improving the plastic properties of Al-Si alloys.

## 1. Introduction

Research on the ways of refining grains of metals and alloys led to the application in industrial practice of methods that ensure grain diameters of several micrometers. Experimental studies indicate a significant improvement in strength properties as a result of grain size reduction [1,2,3].

Materials with a submicron structure show high stability, and grain growth is observed in them at a relatively high temperature, often exceeding 0.4–0.5 of the melting point, despite the increased diffusivity observed when grain size decreases [4,5,6]. It is hypothesized that the stability of such a structure may be determined by a large number of triple boundary points. They have an inhibiting effect on the migration of boundaries or the precipitation process related to the decrease in the solubility of alloy additions during grain growth [7,8].

In the case of non-ferrous metals, efforts are still being made to improve their properties through fine-grained refinement. The advantages of aluminum as a construction material, together with its lightness, are still a strong incentive to search for effective and economically effective methods of improving its strength properties. The goal of this research is also to obtain materials with a grain size of less than one micrometer [9,10,11].

Increasing the strength properties can be effectively obtained by SPD (Severe Plastic Deformation) methods. These methods assume a very high degree of plastic deformation of the material while limiting the possibility of spontaneous recrystallization. These methods consist of introducing a large number of defects into the material as a result of plastic deformation, which, after spatial reorganization and mutual reaction, are able to create a large number of grain boundaries and, thus, grain refining. SPD methods are based on plastic deformation with a very high proportion of the compressive hydrostatic stress state, which aims to prevent loss of material cohesion [12,13,14,15].

One of the methods of producing fine-grained materials is the method of rapid solidification (RS). With a properly selected chemical composition and a sufficiently high cooling rate, an amorphous material (metallic glass) can be obtained [16,17]. Therefore, in this type of material, the rapid solidification method is used mainly to refine the structural components, which leads to a significant increase in strength properties. In the RS method, the speed of heat dissipation during cooling and the heat released during crystallization significantly affect the speed of crystallization. Therefore, it is best to use thin ribbons cast on an intensively cooled copper cylinder. Another solution is to use metallic powders that can be obtained by spraying liquid metal. During the process, these powders are cooled in a shield of inert gas, e.g., argon. In the case of the Spray Deposition method, it is important that the liquid droplets do not crystallize before they hit the surface of the drum [18,19,20,21].

The melt spinning method can be used to obtain high-strength, corrosion-resistant aluminum alloys. The use of this method in the case of AA 5083 alloy resulted in an increase in the tensile strength by 65%, the yield strength by 45%, and the elongation by 14% [22]. The RS process leads to the grain refinement of various materials. It is thought that the RS contributes to the grain refinement of the AZ91 alloy [23]. Rapid-solidification technology can significantly refine Al-Zn-Mg-Cu alloy grains. After extrusion, the tensile strength and elongation of the extruded bar were 466 MPa and 12.9%, respectively. After T6 heat treatment, the tensile strength of the alloy reached 636 MPa, while elongation decreased to 10.5% [24].

Al-Si alloys are among the most commonly used aluminum alloys in automotive applications (e.g., engine components). Silicon significantly affects the strength of Al-Si alloys by transferring the load from the Al matrix to the hard (rigid) Si phase in the microstructure (load capacity). Casting parameters (i.e., solidification rate, element segregation), as well as the size and distribution of microstructural components in Al-Si alloys (i.e., Si particle morphology, intermetallic compounds, spacing between secondary dendrites) have a direct impact on the microstructure, mechanical properties, and behavior of the material in case of failure (or cracking) [25,26,27,28,29].

This paper presents an analysis of cast Al alloys with different silicon contents (5%, 11%, and 20%), which were obtained by two methods: gravity-casting and extrusion, as well as rapid solidification and plastic consolidation in the extrusion process. The choice of methods was dictated by the discrepancies in properties that arise during the production of materials by these methods. These differences result directly from the rapid solidification process. The literature lacks information on the comparison of the properties of this type of alloy and the impact of the melt spinning method on the fragmentation of the silicon phase and obtaining a morphology close to spherical.

## 2. Materials and Methods

Aluminum alloys with different silicon contents were used for the tests. The chemical composition of the starting materials is shown in Table 1. First, these materials (Table 1) were gravity-cast into an ingot with a diameter of 38.2 mm and a height of 60 mm. The charges were extruded at a temperature of 375 °C using an extrusion speed of 1 mm/s and processing λ = 25 to form rods with a diameter of 8 mm. A 100 T press manufactured by HYDROMET (Bytom, Poland) was used for extrusion.

The second set was cast in the process of rapid solidification (RS) using the melt spinning method, pressed, and hot extruded. The materials were inductively melted in a graphite crucible in an argon protective atmosphere and then cast onto a copper rotating cylinder rotating at a circumferential speed of 10 m/s. This technique allows for an alloy cooling rate of 106 K/s. Rapidly solidified ribbons with a thickness of 30–70 µm and a width of about 2.5 mm were compacted into a briquette with a diameter of 38 mm and a height of 10 mm. The rapid solidified ribbons and briquette for AlSi5 RS is shown in Figure 1.

RS compaction of the ribbons was carried out at ambient temperature on the KHPES 100 Georg KIRSTEN D-54427 Kello press under a pressure of 100 bar. Six briquettes were made for each rapid-solidified material, which constituted the charge for the extrusion process. The extrusion process was carried out with the above-mentioned parameters.

Samples were taken from the extruded rods, and microsections were prepared for microstructure observation. The samples were ground on abrasive papers using 240–800 grits (paper grits) and then polished with diamond pastes (DP-Suspension P from Struers) with grits of 9, 3, and 1 µm. Finishing polishing was carried out using a colloidal suspension of silica OP-S from Struers. Grinding and polishing were performed on a RotoPol 11 device (manufactured by Struers, Copenhagen, Denmark). Microstructure observations were performed using an Olympus GX51 light microscope (Olympus Inverted Metallurgical Microscope, Olympus, Tokyo, Japan) and a Hitachi SU-70 scanning electron microscope (Hitachi High-Technologies Corporation, Tokyo, Japan). EDS spectroscopy was used for the study of an element’s distribution. The shape ratio was calculated using ImageJ (Rockville, Bethesda, MD, USA). Circularity was used as the shape factor. The static tensile test was carried out at an ambient temperature in accordance with PN-EN ISO 6892-1:2020-05 [30] using a Zwick Roel Z050 testing machine (manufactured by ZwickRoell Group, Ulm, Germany). From the beginning, middle, and end of each rod, cylindrical samples were taken and made in order to determine the mechanical properties in the tensile test. Samples with a diameter of 6 mm and a length of the measuring base of 30 mm were deformed at a speed of 8 × 10^−3^ s^−1^. The high-temperature compression test was carried out using the MTS 880 testing machine (MTS Systems Corporation, Eden Prairie, MN, USA). Samples with dimensions of 8 mm in diameter and 11 mm in length were used for the compression test. The samples were cut directly from the rod after the extrusion process. The Vickers hardness measurement was carried out with a load of 19.61 N using a Shimadzu HMV-2 T device microhardness tester (Shimadzu Corporation, Kyoto, Japan).

## 3. Results and Discussion

Traditional foundry methods of grinding grains of metallic materials, unfortunately, do not allow for sufficient particle and grain size. Figure 2 shows a longitudinal section of ribbons with different silicon contents. All ribbons are approximately 40 µm thick. There are visible differences in the microstructure on the side of the cylinder (bottom of the picture) and on the side of the air (top of the picture). From the side of the cylinder, crystals of a solid solution of aluminum are visible, arranged orthogonally to the direction of heat dissipation, with precipitates of phases that are rich in silicon. From the air side, a typical dendritic structure can be seen, which indicates a slower heat dissipation from this side. Similar results can be found in the literature [22,31] In addition, from the observation of the microstructures of the strips, it can be seen that, depending on the amount of silicon, the zones of orthogonally growing crystals have different sizes. In the case of the AlSi5 RS material, this zone was 20 µm; for the AlSi11 RS material, it was 11 µm; and in the case of the AlSi20 RS material, this zone was only 6 µm. In addition, in the zone on the air side, an increase in grain size can be seen with increasing silicon content in the material.

Long acicular silicon crystals with sharp corners and edges irregularly occurring in the Al-Si alloys (Figure 3A,C,E) have a very negative effect on the mechanical properties and sometimes completely degrade the mechanical workability of these materials. The use of rapid solidification (RS) and plastic consolidation technology radically changes the type of grain structure of the material. The products of this technology are materials with a very fine, equiaxed structure of primary silicon, with grain sizes of several micrometers (Figure 3B,D,F).

The analysis of the size and shape factors of silicon particles in the tested materials is shown in Figure 4 and Figure 5. The average silicon particle size increases significantly with increasing Si content. A significant discrepancy in the Si particle size for AlSi20 can also be seen, which is also confirmed by the determined shape factor for this material (Figure 5). However, in the case of materials produced by rapid solidification, we observe a comparable size of Si particles for all RS materials (Figure 4), and the shape of these particles is close to spheroidal (Figure 5).

Particle refinement after the RS process is well presented in the element distribution map (Figure 6, Table 2, Table 3 and Table 4). Figure 7A–C show the microstructures of materials with 5%, 11%, and 20% silicon content. In the figures, silicon particles with a size of several nanometers to about 2 µm can be seen. These particles are evenly distributed in the material. Not only the Si particles are refined, but also the Al-Si-Fe phase, which also changed its shape from irregular with sharp edges to regular and spherical. In the case of rapid solidification materials, silicon crystals, without a noticeable separation into primary and eutectic silicon crystals, surround grains of the aluminum-rich phase with a size of 1 to 4 µm (Figure 7). Table 5 presents the results of grain size measurements for the tested materials. The material containing 20% Si has the largest grain. For the AlSi20 sample, the grain size is 9.87 µm, and after the rapid solidification process (AlSi20 RS), the grain was reduced to 3.78 µm. The smallest grain size is characterized by a sample with a silicon content of 5% (for AlSi5, it was 5.9 um, while after the rapid solidification process (AlSi5 RS), it is fine-grained at 1.38 µm). In general, the results of the microstructure tests show that the rapidly solidified and plastically consolidated material has a refinement structure that is thermally stable and does not undergo significant changes during high-temperature deformation (hot extrusion).

It is worth emphasizing that the temperature of plastic consolidation of rapidly crystallized strips significantly affects the maintenance of fragmentation of microstructure components. Too low a temperature hinders diffusion processes and limits plastic consolidation to the mechanical joining of metallic particles. Then the strength properties are low compared to those expected as a result of the fragmentation of the microstructure. On the other hand, too high a consolidation temperature causes significant grain growth and leads to undesirable coagulation of precipitates and changes in particle morphology. During extrusion in the plastic flow zone, cracking of the external oxide coatings on the surface of rapidly solidified ribbons occurs, which leads to the exposure of the proper surface of the material. This facilitates the adhesive connection of particles and increases the efficiency of diffusion through the boundaries of particle joining [32,33,34].

Figure 8 and Table 6 show the results of the tensile test at room temperature for rapid solidification materials. The results were compared with the properties of gravity-cast and hot-extruded materials. The rapid solidification process significantly increases the strength properties of all tested materials. The highest YTS value was obtained for AlSi20RS, and it is an almost two-fold increase in strength properties (Table 6). A similar trend is observed for hardness measurements using the Vickers method (HV2). The lowest hardness value is obtained by AlSi5 RS (HV2 48), and with the increase in the silicon content and the increase in the amount of crushed particles, the hardness of the consolidated materials increases (Table 6). It is worth emphasizing, however, that all alloys, hypoeutectic (AlSi5 RS), eutectic (AlSi11 RS), and hypereutectic (AlSi20 RS), are also characterized by higher elongation compared to gravity-cast and extruded materials. Table 6 also shows the density results of all tested samples. In the case of the alloy with 5 and 11% silicon, the density of rapid solidified samples is the same as that of gravity-cast samples. For these materials, no porosity was observed during microstructural observations. The density of AlSi20 RS is slightly higher than that of AlSi20. This is due to the fact that in the case of the gravity-cast AlSi20 alloy, porosity could be seen in the material, which appeared in the vicinity of large Si particles that probably cracked during the extrusion process. In the case of AlSi20 RS, the Si particles were refined during the melt spinning process, and we do not observe porosity in the extruded material.

Figure 9 shows the fractures of the samples after the tensile test. The observations indicate that in the case of gravity-cast and extruded materials, the lower elongation values are due to the presence of large silicon particles, which are marked with arrows in Figure 9. In addition, smaller particles located at the bottom of the microwells can be noticed. Sharp edges testify to the initiation of the cracking process on these particles. The samples after the rapid solidification process have a significantly different fracture morphology. Fractures in these samples, regardless of the amount of silicon, have the character of plastic cracking, which is confirmed by the elongation values obtained in the tensile test. Numerous micro holes with very fine particles of the second phase are located on the entire surface of the fracture. The micro holes created after the tensile test are, in most cases, spherical in shape and of similar size throughout the fracture. The second-phase particles are much smaller in the RS material compared to the fractures obtained after gravity-casting and extrusion.

Examples of the effects of increased temperature on the mechanical properties of rapidly solidified AlSi alloys are shown in Figure 10. The mechanical properties of extruded samples were evaluated in the range 20–500 °C using compression tests. The rapidly solidified materials are characterized by much higher strength parameters in comparison to materials cast by gravity and plastically consolidated. These materials show a significant ability to undergo plastic deformation at high temperatures.

When designing alloys with a higher content of silicon, e.g., materials intended for heavily loaded combustion engine pistons, special attention should be paid to the problems associated with the lack of a sufficiently effective method of modifying the morphology and size of primary silicon crystals in alloys with hypereutectic composition [35,36,37]. Long, acicular silicon crystals have a very adverse effect on the mechanical properties, can contribute to numerous cracks, and sometimes completely degrade the mechanical workability. These types of materials can be modified, for example, with phosphorus, which improves machinability in some hypereutectic materials but does not adequately improve strength. The addition of Ni, Fe, or Cu improves the strength properties at high temperatures but strongly deteriorates the casting properties. This leads to porosity and cracking in castings and completely deprives such materials of the ability to undergo plastic deformation [38,39,40]. Melt-spinning casting of such materials into thin ribbons gives a chance to better adjust the properties of the material to the application, provided that the process of plastic consolidation of dispersed alloy forms is effectively carried out.

Figure 11A shows the yield strength depending on the temperature for the gravitationally cast and extruded material, while Figure 11B shows research for samples after the melt spinning process. The addition of silicon to gravity-cast materials does not significantly increase the yield point. A different situation can be observed in the case of samples after rapid solidification. Strong fragmentation of silicon particles and grains resulted in increased strength properties. As shown in Figure 7, in the samples after the rapid solidification process, silicon particles have sizes ranging from several nanometers to several micrometers. The hot pressing process causes the accumulation of defects in the material, which results from the simultaneous impact of plastic deformation and temperature. During the compression test, a recovery process is triggered, which leads to a reduction in the internal energy in the material due to dislocation displacement by climbing and sliding. The climb of dislocation often occurs in metals with high stacking fault energy, of which aluminum is an example. This leads to pinning the dislocation by fine silicon particles that prevent it from moving. Further recovery can occur through polygonization processes consisting of the movement of dislocations and the formation of parallel rows in places of greater stress. This leads to the formation of small-angle boundaries (subgrains). In the material obtained by gravity-casting, the silicon particles are much larger and there are much fewer of them, which means that the strength properties after the compression test are much lower than in the case of samples after rapid solidification [41,42,43].

## 4. Conclusions

Rapid solidification by the melt-spinning method is an effective method of fragmentating structural components in Al-Si alloys. The advantage of these materials is the homogeneity and high stability of the morphology of the precipitates in a wide temperature range, which allows the use of this type of material in the production of products that can be used at elevated temperatures.

Rapidly solidified Al-Si materials are characterized by a comparable size of Si particles, regardless of the silicon content, and the shape of these particles is close to spheroidal. Not only Si particles are fragmented, but also the Al-Si-Fe phase, which also changed its shape from irregular with sharp edges to regular and spherical.

The mechanical properties of materials obtained by combining the technologies of rapid crystallization and plastic consolidation in the extrusion process are significantly higher in comparison to gravity-cast and extruded materials. An increase in strength properties was found in the case of the AlSi5 RS alloy by 20%, in the case of AlSi11RS by 25%, and in the case of the alloy containing 20% Si by as much as 86% in relation to gravity-cast and extruded alloys.

## Figures and Tables

**Figure 1 materials-16-05223-f001:**
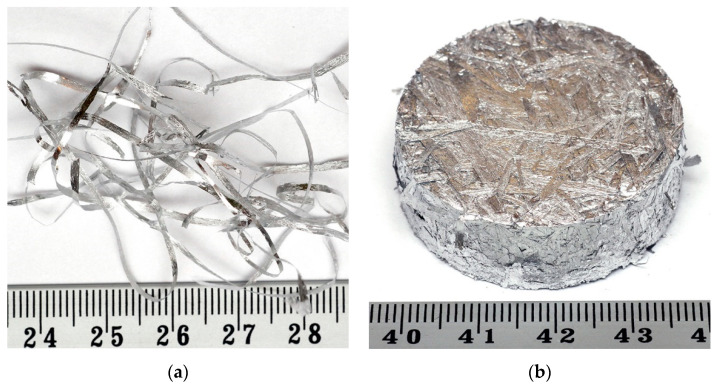
AlSi5 RS: ribbons (**a**) and briquette (**b**).

**Figure 2 materials-16-05223-f002:**
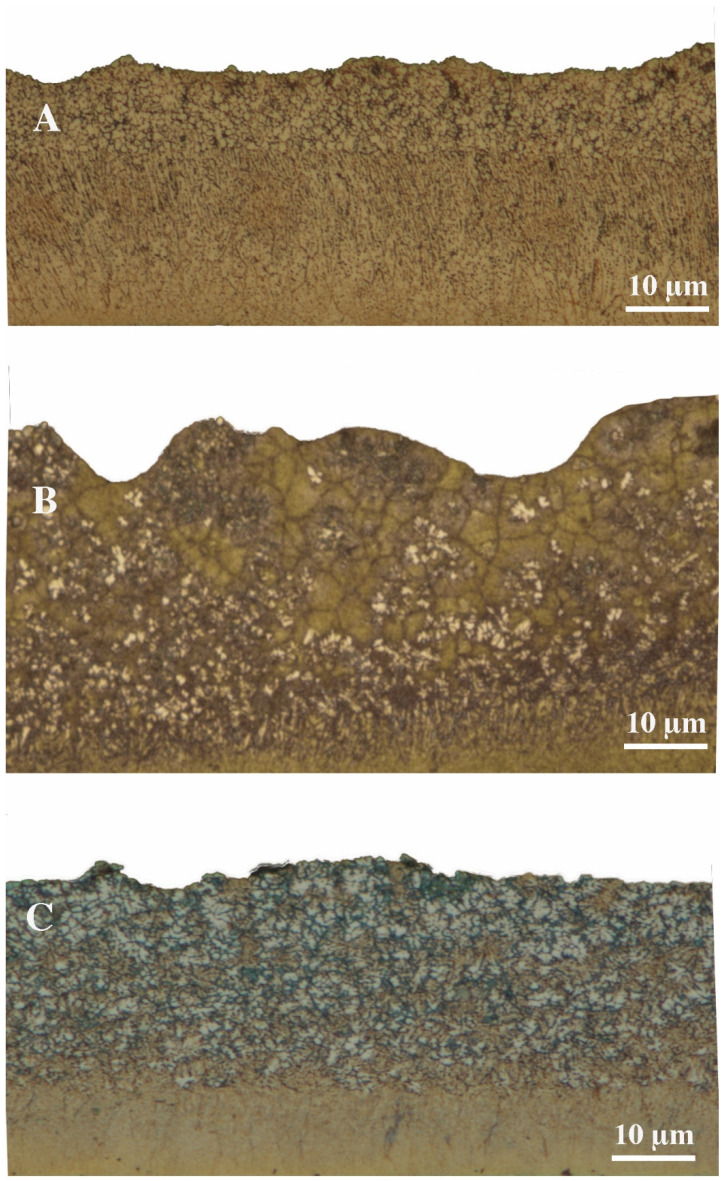
The microstructure of the ribbons after the melt spinning process: (**A**) AlSi5 RS, (**B**) AlSi11 RS, and (**C**) AlSi20 RS.

**Figure 3 materials-16-05223-f003:**
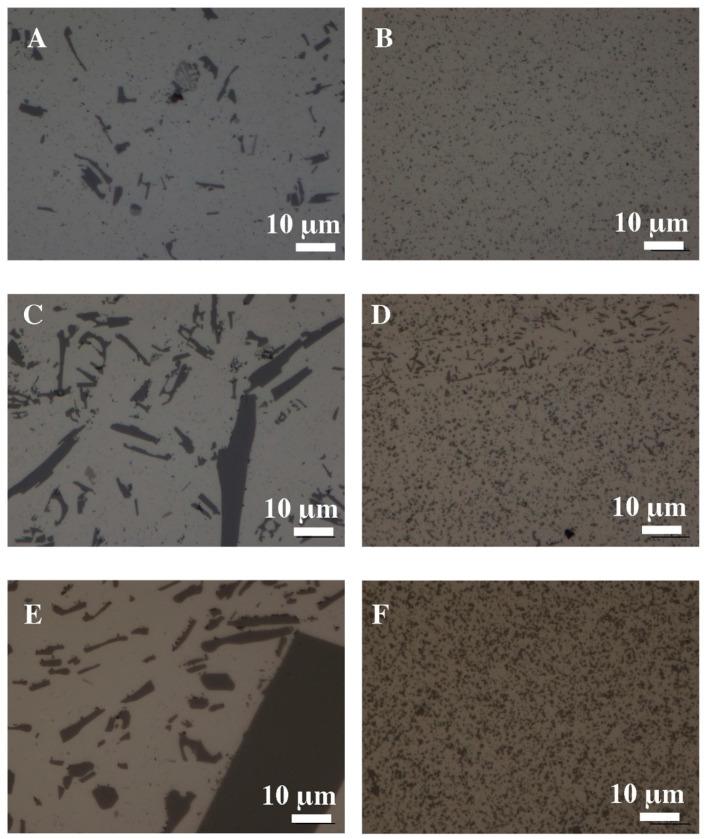
The microstructure of Al-Si alloys after the extrusion process: (**A**) AlSi5, (**C**) AlSi11, (**E**) AlSi20, (**B**) AlSi5 RS, (**D**) AlSi11 RS, (**F**) AlSi20 RS, light microscopy.

**Figure 4 materials-16-05223-f004:**
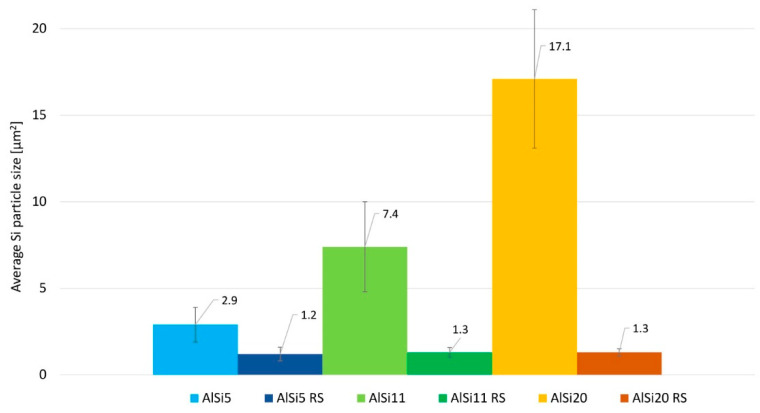
Average Si particle size in Al-Si alloys.

**Figure 5 materials-16-05223-f005:**
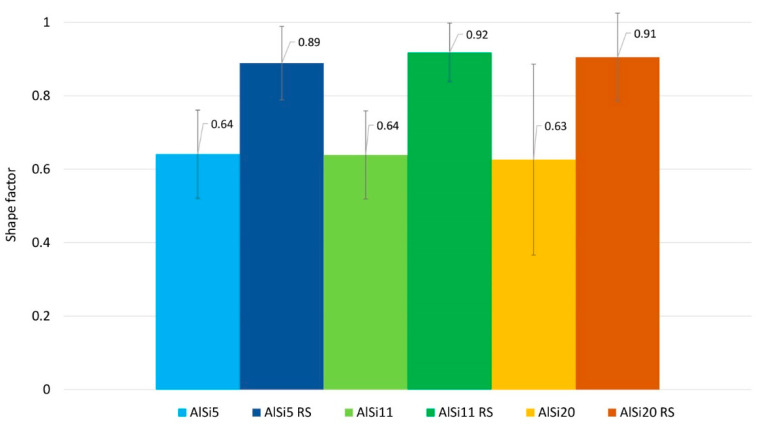
Shape factor distribution of Si particles in Al-Si alloys.

**Figure 6 materials-16-05223-f006:**
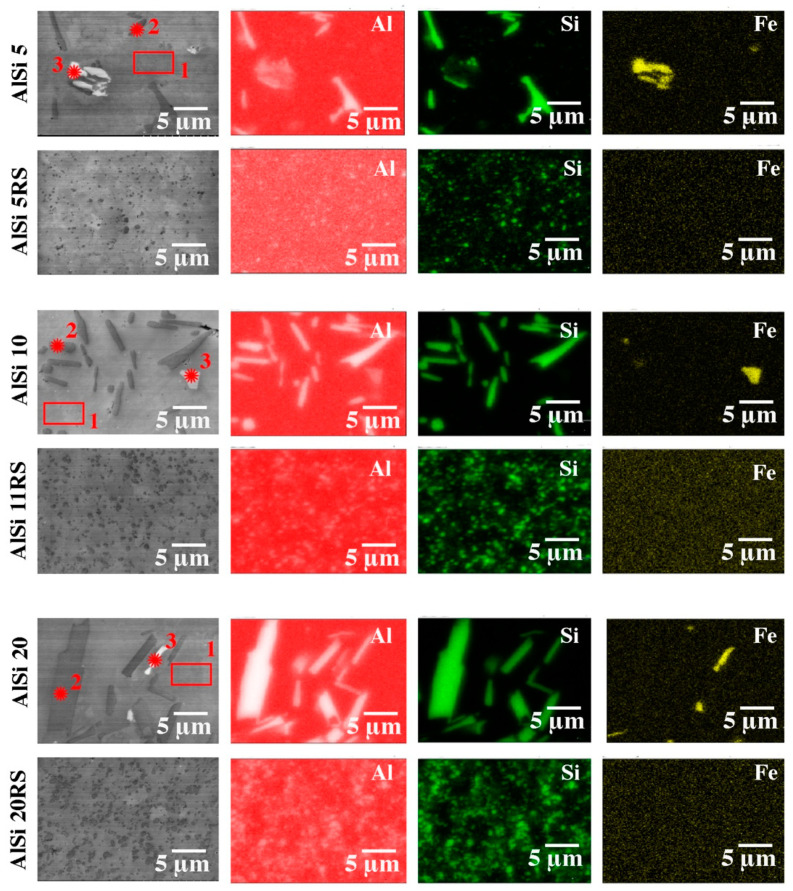
The microstructure of Al-Si alloys after the extrusion process: SEM with the EDS chemical mapping of components is shown in Table 2, Table 3 and Table 4.

**Figure 7 materials-16-05223-f007:**
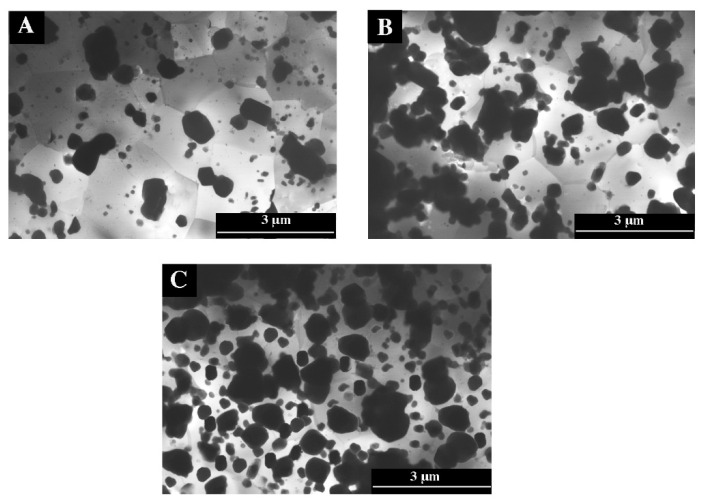
The microstructure of RS alloys after the extrusion process, TEM. (**A**) AlSi5 RS, (**B**) AlSi11 RS (**C**)AlSi20 RS.

**Figure 8 materials-16-05223-f008:**
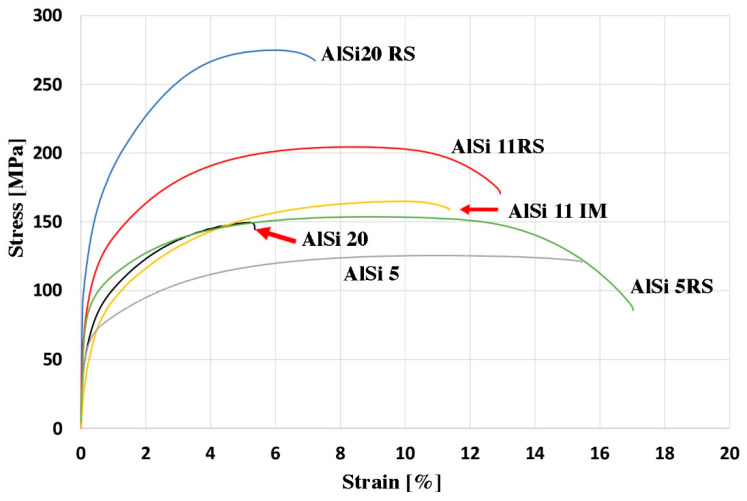
Stress-strain curves of Al-Si alloys after the extrusion process are deformed by a tension test.

**Figure 9 materials-16-05223-f009:**
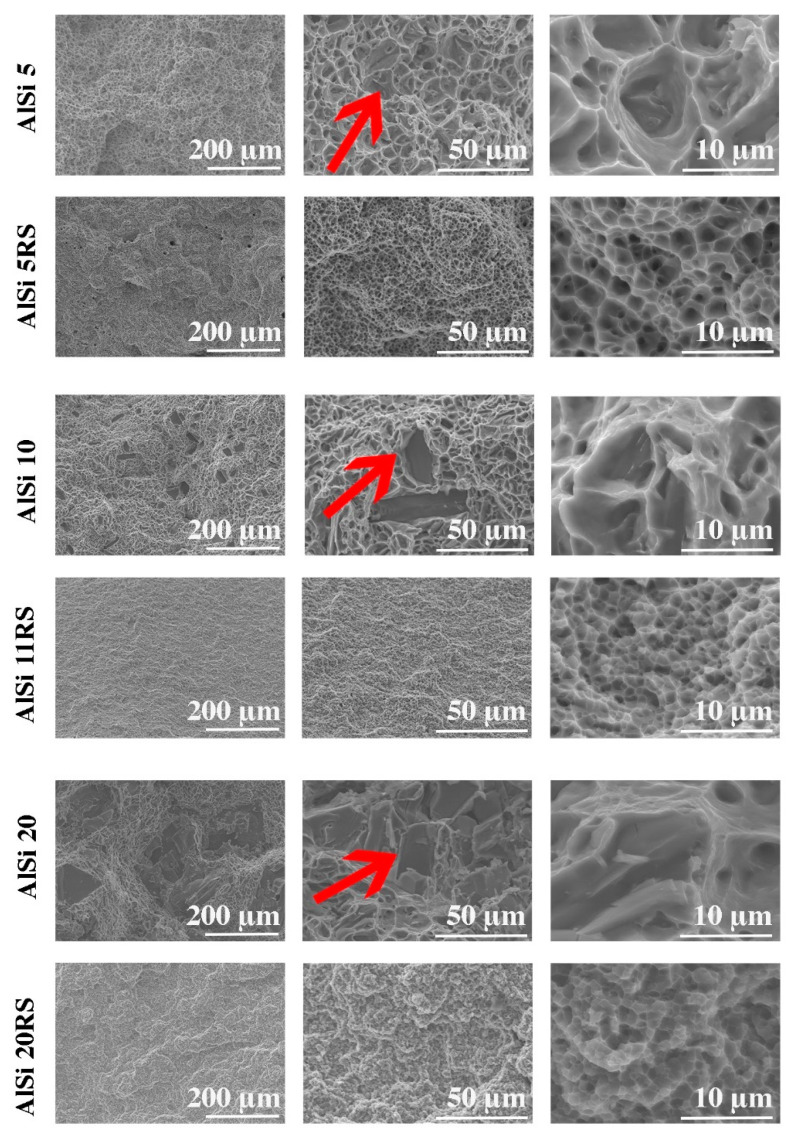
Fracture topography of the tested materials.

**Figure 10 materials-16-05223-f010:**
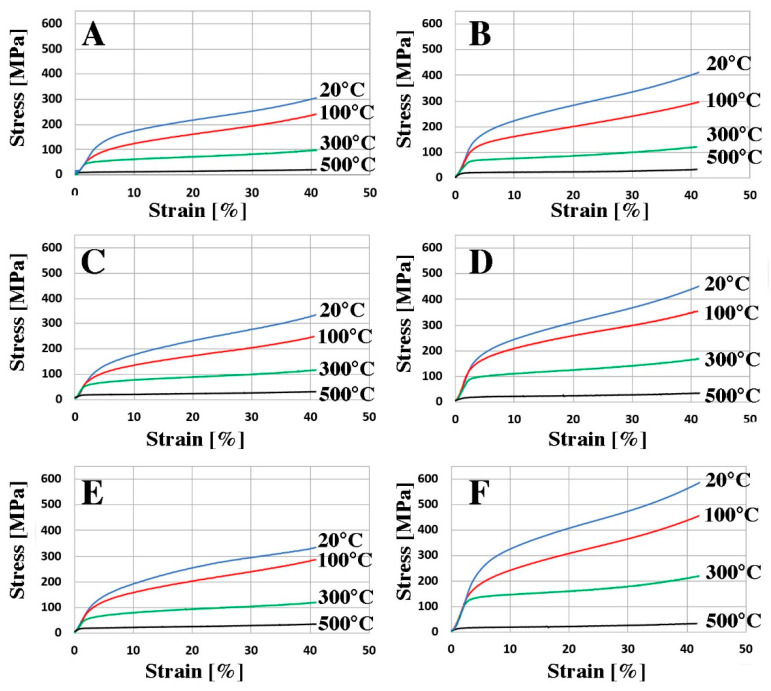
Stress-strain curves received for as-extruded materials deformed by compression at a constant true strain rate. (**A**) AlSi5, (**C**) AlSi11, (**E**) AlSi20, (**B**) AlSi5 RS, (**D**) AlSi11 RS, (**F**) AlSi20 RS. Deformation temperature is marked in the figure.

**Figure 11 materials-16-05223-f011:**
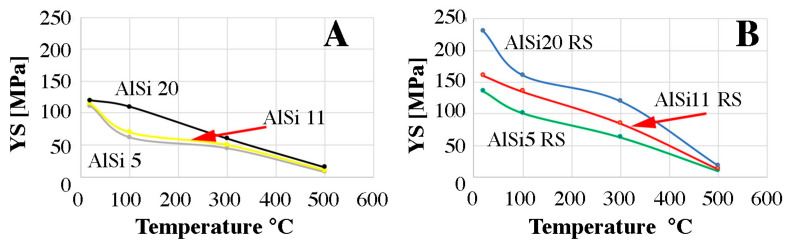
Yield strength as a function of the temperature of the tested materials. (**A**) gravity casting (**B**) RS alloys.

**Table 1 materials-16-05223-t001:** Chemical composition of the Al-Si alloy.

Element	Al	Si	Fe	Cu	Mg	Mn	Other
AlSi5	94.28	5.23	0.17	0.03	0.07	0.02	0.20
AlSi11	88.26	11.41	0.16	0.01	0.02	0.03	0.13
AlSi20	79.37	20.21	0.17	0.02	0.04	0.03	0.16

**Table 2 materials-16-05223-t002:** Results of X-ray chemical analysis (EDS) for *1, *2, and *3 marked in Figure 6 (AlSi5).

AlSi5	Mg-K	Al-K	Si-K	Mn-K	Fe-K	Cu-K
1	0.00	97.00	2.99	0.00	0.10	0.00
2	0.00	8.96	91.04	0.00	0.00	0.00
3	0.00	55.32	28.60	0.16	15.92	0.00

**Table 3 materials-16-05223-t003:** Results of X-ray chemical analysis (EDS) for *1, *2, and *3 marked in Figure 6 (AlSi11).

AlSi11	Mg-K	Al-K	Si-K	Mn-K	Fe-K	Cu-K
1	0.00	97.89	2.01	0.00	0.10	0.00
2	0.00	7.86	92.14	0.00	0.00	0.00
3	0.00	54.32	29.80	1.16	14.72	0.00

**Table 4 materials-16-05223-t004:** Results of X-ray chemical analysis (EDS) for *1, *2, and *3 marked in Figure 6 (AlSi20).

AlSi20	Mg-K	Al-K	Si-K	Mn-K	Fe-K	Cu-K
1	0.00	97.91	1.99	0.00	0.10	0.00
2	0.00	8.76	91.24	0.00	0.00	0.00
3	0.00	56.32	28.70	0.16	14.82	0.00

**Table 5 materials-16-05223-t005:** Average grain diameter of the researched materials.

	Average Grain Diameter	Standard Deviation
AlSi5	5.90	0.73
AlSi5 RS	1.38	0.15
AlSi11	6.16	0.64
AlSi11 RS	1.67	0.22
AlSi20	9.87	1.37
AlSi20 RS	3.78	0.16

**Table 6 materials-16-05223-t006:** Properties of Al-Si alloys.

Element	UTS, MPa	YS, MPa	Elongation, %	Hardness, HV2	Density, g/cm^3^
AlSi5	131	81	17.1	35	2.67
AlSi5 RS	155	108	15.2	48	2.67
AlSi11	162	90	11.1	45	2.64
AlSi11 RS	203	132	12.6	64	2.64
AlSi20	148	94	4.1	50	2.56
AlSi20 RS	276	168	6.2	86	2.61

## Data Availability

Not applicable.

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
