# Peer review of "Microstructure and Mechanical Properties of Al-Si Alloys Produced by Rapid Solidification and Hot Extrusion"

_materials, 2023, doi:10.3390/ma16155223_

Round 1

Reviewer 1 Report

1.       The relevant research methods are not detailed enough in the section on experimental materials and methods. For example, preheating temperature and time of the alloy, extrusion rate, and process parameters for rapid solidification, etc. The test parameters for the mechanical properties, and the size of the samples need to be given as well.

2.       The AlSi20 RS sample has the best performance, and it is suggested to give the corresponding TEM images and its diffraction patterns.

3.       In Fig. 2, it is not easy to distinguish the grain size of the alloys, which has an essential impact on the analysis of the mechanical properties of the alloys, and it is recommended to supplement the relevant images (TEM or EBSD).

4.       The mechanism of the effect of rapid solidification and silicon content on the high-temperature mechanical properties of the alloys needs to be discussed in depth, preferably with quantitative calculations or a physical model.

5.       The abstract should present in a concise manner the main outcome of the paper. The current abstract should be polished along this line. Please bear in mind that a sharp abstract is central for article visibility.

6.       The conclusion should summarize the main findings and clarify how the results presented in the paper have made the field progressing, for instance in comparison with other works. A sole summary of the findings is not sufficient. Please polish the conclusion along this view.

7.       The references are poorly formatted, and there is a suggestion to cite some of the newly published papers, such as some of the recent relevant papers in this journal.

Please carefully go through the manuscript to polish the language, including abstract.

Author Response

Dear Reviewer,

The Answers is in the attach

Reviewer 2 Report

In this work, the microstructure and mechanical properties of Al-Si alloys with different Si contents were evaluated. In more detail, one set of samples were fabricated by casting and subsequent extrusion. Another set of samples were fabricated by melt spinning, compressed to form bulk samples and then extruded. Please find my comments below:

1) Introduction is very brief, generic and doesn`t explain the significance of this work. Please expand the introduction and include works in Al based alloys that were fabricated and processed with rapid solidification and extrusion. Discuss the microstructure and mechanical properties of those alloys. At the end of the introduction explain the novelty of this work.

2) In the introduction the authors mention grain refinement. What is the grain size of the two different sets of samples?

3) In materials and methods the two different approaches to fabricate Al-Si alloys are not properly described. To my best understanding, the first part of samples were cast and then extuded. The second set were melt spun, compressed and then extruded. Please explain the process in more detail.

4) The authors include in the results parts that are generic/vague and belong to the introduction like 99-101, 107-116. Please remove that and focus in commenting the results.

5) What is the microstructure of the RS samples before extrusion? A few SEM images and/or XRDs would help to improve the discussion.

6) What is the porosity of the samples? Especially for the RS samples that were compresed this is of high importance.

7) Why in Al-11Si and Al-20Si elongtion is improved for the RS samples over the conventinally fabricated ones while in Al-5Si elongation decreases? Explain in more detail why this is happening.

8) In lower temperatures (Fig 8) RS appears to have a more significant effect on the mechanical properties as compared to the higher temperatures. Provide an explanation and enrich the discussion please.

9) Conclusions must be improved. Explain briefly the novelty of this work, the experimental procedure and the most important findings.

Author Response

(The authors gave the same response as above.)

Round 2

Reviewer 1 Report

I carefully re-evaluated the revised manuscript and the author's response to the reviewer's comments. The authors have addressed all the concerns raised by and manuscript quality has been increased by the inclusion of suggested studies/results. I recommend the publication of the manuscript in its current format.

Reviewer 2 Report

Authors implemented all the requested changes. The manuscript is improved and is ready for publication.